# Hemodynamic and Respiratory Changes following Prone Position in Acute Respiratory Distress Syndrome Patients: A Clinical Study

**DOI:** 10.3390/jcm12030760

**Published:** 2023-01-18

**Authors:** Maria Baka, Dimitra Bagka, Vasiliki Tsolaki, George E. Zakynthinos, Chrysi Diakaki, Konstantinos Mantzarlis, Demosthenes Makris

**Affiliations:** 1Critical Care Department, University Hospital of Larissa, 41111 Larissa, Greece; 2Third Cardiology Clinic, Sotiria Hospital, 11527 Athens, Greece; 32nd Critical Care Department, Attikon University Hospital, Medical School, National and Kapodistrian University of Athens, 11527 Athens, Greece

**Keywords:** acute respiratory distress syndrome, prone positioning, Oxygenation Index, shunt fraction, pulmonary pressure, pulmonary vascular resistances

## Abstract

Background: Limited data are available for the oxygenation changes following prone position in relation to hemodynamic and pulmonary vascular variations in acute respiratory distress syndrome (ARDS), using reliable invasive methods. We aimed to assess oxygenation and hemodynamic changes between the supine and prone posture in patients with ARDS and identify parameters associated with oxygenation improvement. Methods: Eighteen patients with ARDS under protective ventilation were assessed using advanced pulmonary artery catheter monitoring. Physiologic parameters were recorded at baseline supine position, 1 h and 18 h following prone position. Results: The change in the Oxygenation Index (ΔOI) between supine and 18 h prone significantly correlated to the concurrent change in shunt fraction (r = 0.75, *p* = 0.0001), to the ΔOI between supine and 1 h prone (r = 0.73, *p* = 0.001), to the supine acute lung injury score and the OI (r = −0.73, *p* = 0.009 and r = 0.69, *p* = 0.002, respectively). Cardiac output did not change between supine and prone posture. Moreover, there was no change in pulmonary pressure, pulmonary vascular resistances, right ventricular (RV) volumes and the RV ejection fraction. Conclusions: The present investigation provides physiologic clinical data supporting that oxygenation improvement following prone position in ARDS is driven by the shunt fraction reduction and not by changes in hemodynamics. Moreover, oxygenation improvement was not correlated with RV or pulmonary circulation changes.

## 1. Introduction

Prone positioning during mechanical ventilation (MV) is used in patients suffering from acute respiratory distress syndrome (ARDS) because it is a strategy that increases significantly the ratio of arterial oxygen partial pressure to the fraction of inspired oxygen (PaO_2_/FiO_2_) in the majority of patients, having a positive effect on their survival and a net benefit for those who are severely hypoxemic [1,2,3,4]. The theory that oxygenation improvement is due to shunt fraction reduction has been supported by many experimental studies, because prone position promotes homogeneous lung aeration in ARDS, while it has no major impact on the regional distribution of blood in the lung [5,6,7]. However, clinical data on this issue are limited. Moreover, it is unclear whether prone position induces significant hemodynamic changes in patients already ventilated with protective protocols and whether these changes are related to the oxygenation improvement after long-term prone positioning [8,9,10].

In the present prospective observational study, we aimed to assess the pathophysiologic changes that occur in the pulmonary vasculature and the circulatory system in mechanically ventilated patients with ARDS who are proned for 18 h and to identify the parameters associated with oxygenation improvement.

## 2. Materials and Methods

### 2.1. Patient Population

The study was conducted in the intensive care unit of a tertiary teaching hospital (12 beds). Patients were included if they fulfilled the following criteria: (a) diagnosis of ARDS [11]; (b) monitored by a pulmonary artery catheter (PAC); (c) stable condition in terms of vital signs and hemodynamics for at least 2 h; and (d) decision to prone the patient (PaO_2_/FiO_2_ < 150 on FiO_2_ > 60% on PEEP = 5 cm H_2_O). Exclusion criteria were: (a) age < 18 years; (b) cardiogenic pulmonary edema; (c) clinical conditions contraindicating prone positioning, i.e., intracranial hypertension, unstable spinal trauma, severe hemodynamic instability; (d) decision to withdraw medical support; (e) cardiac rhythm other than sinus rhythm (in order to avoid miscalculations in the right ventricular diastolic volume (EDV); (f) chronic right ventricular (RV) failure; and (g) increased intra-abdominal pressure. Decisions regarding the placement of a PAC were taken by the treating physicians, whereas all cases were discussed in a daily multidisciplinary meeting.

### 2.2. Outcomes

The primary outcome in this study was the change in the Oxygenation Index (ΔOI_S18hP_) between the values obtained in the baseline semi recumbent position (SR_BAS_), i.e., supine position, and at 18 h of pronation (18hProne). The Oxygenation Index is defined as OI = FiO_2_ × Mean Airway Pressure/PO_2_. We assessed the association between the ΔOI_S18P_ and hemodynamic/respiratory variables using the values at SR_BAS_ at the first hour of prone position (1hProne) and their respective changes over 18 h at prone posture.

### 2.3. Settings

Patients were sedated and placed in the 45-degree SR_BAS_ and ventilated with conventional lung-protective mechanical ventilation (Evita 4 Drager Medical, Lubeck, Germany) for a period of at least 2 h to obtain a steady state (optimization period). Specifically, initial PEEP and Tidal Volume (TV) titration were based on the ARDSnet protocol principles [12]; physicians were advised to use cardiac ultrasound data for subsequent titration in order to avoid Acute Cor Pulmonale (RVEDA/LVEDA = right ventricular end-diastolic area/left ventricular end-diastolic area ratio > 0.6 and/or interventricular septum paradoxic motion). Respiratory rate was set to achieve PaCO_2_ 40–90 mmHg and pH > 7.15, at a maximum set respiratory rate of 35 bpm. Consequently, in the case of a change in the need for oxygenation and/or ventilation during the prone position, FiO_2_ was allowed to be titrated first and, secondarily PEEP, in order to achieve PaO_2_: 55–80 mmHg or SpO_2_: 88–95%. Patients were ventilated in the prone posture for at least 18 h and were then turned to supine position. Supination after PP was performed on a time schedule, unless clinical or physiologic deterioration occurred, dictating interventions in the supine position (e.g., inserting a new intravenous catheter for mechanical renal replacement therapy). Scheduled measurements were evaluated at SR_BAS_ (baseline), at 1hProne, and 18hProne. A Swan-Ganz Continuous-Cardiac-Output/End-Diastolic Volume/Thermodilution Catheter (7.5-Fr, Edward’s Lifesciences, Unterschleissheim Germany) and disposable pressure transducers were used. Calculation of the shunt fraction ($\dot{Q}$s/$\dot{Q}$t) was done using the shunt equation, known as the Berggren equation [13]; the ARDS score or Acute Lung Injury (ALI) score was calculated as suggested previously [14] (for more details see additional information in Appendix A).

The protocol was approved by the local institution review board (1/22032002). Patients’ next of kin were informed. The requirement for written informed consent was waived.

### 2.4. Statistical Analysis

Analysis was performed on an intention-to-treat basis. Data are presented as mean ± SD, unless otherwise indicated. Data between supine and prone position were compared using the Fisher’s exact test for categorical variables and the *t*-test or the Mann–Whitney test as appropriate for continuous variables. Changes between supine and prone position for each patient were assessed by the paired *t*-test or the Wilcoxon test, as appropriate. Correlations between outcome variables and clinical or physiologic variables were assessed appropriately either by Pearson’s r or by Spearman’s rho. To determine the prognostic value of various parameters in distinguishing patients who present an improvement in the ΔOI_S18hP_ (above the mean value), receiver operating characteristic (ROC) curves were constructed to assign cut-off values and their diagnostic utility. All statistical tests were two-sided. A result was considered statistically significant when *p* < 0.05. Analysis was performed using SPSS v.20 statistical software for Windows.

## 3. Results

A total of 18 subjects entered the study. The patients’ clinical data are presented in Table 1, whereas the hemodynamic and pulmonary function data at SR_BAS_, 1hProne, and 18hProne are presented in Table 2. The change from SR_BAS_ to 18hProne position was significantly associated with improvements in the OI and PO_2_/FiO_2_.

### 3.1. Oxygenation Changes and Their Determinants

The ΔOI_S18P_ was significantly related to several hemodynamic and oxygenation indices (Table 3).

The highest correlations were found between the ΔOI_S18P_ and the respective change in $\dot{Q}$s/$\dot{Q}$t (Δ$\dot{Q}$s/$\dot{Q}$t_S18P_) (r = 0.75, *p* = 0.0001) (Figure 1).

The Δ$\dot{Q}$s/$\dot{Q}$t_S18P_, in turn, was significantly correlated with the respective changes in respiratory system compliance (r = −0.60, *p* = 0.01) (Figure 2), driving pressure (r = 0.51, *p* = 0.02), and oxygen consumption VO_2_ (r = −0.50, *p* = 0.03).

The ΔOI_S18P_ also correlated with the Oxygenation Index at SR_BAS_ [OI_SRBAS_ (r = 0.69, *p* = 0.002)] and the ARDS score at SR_BAS_ (r = 0.73, *p* = 0.009) (Table 3). Moreover, the ΔOI_S18P_ was significantly associated with the change in the Oxygenation Index between SR_BAS_ to 1hProne (ΔOI_S1P_) (r = 0.73, *p* = 0.0001).

The ΔOI_S1P_ (cut-off at −1.75) demonstrated remarkable sensitivity and specificity (100% and 77.8%, respectively, Likelihood ratio 4.5) in identifying patients who presented a positive response in the ΔOI_S18P_ following prone positioning (Area Under the Curve 0.93, *p* = 0.002, Figure 3).

Significant relationships between the Oxygenation Index at 18hProne (OI_18p_) and clinical, hemodynamic and respiratory indices are presented in Appendix A).

### 3.2. Hemodynamic Changes

Hemodynamic changes at SR_BAS_, 1hProne, and 18hProne are shown in Table 2. Interestingly, pulmonary artery pressures and pulmonary vascular resistance (PVR) did not change, although there was a significant improvement in oxygenation. Moreover, there was no significant change in cardiac output (CO) between SR_BAS_ and any of the two time-points in the prone posture. In addition, the RV end-diastolic volume and the RV ejection fraction did not change either, as probably expected, as the PVR and CO had not changed. However, we found that the change in the CI between supine and 1hProne correlated (r = 0.48 *p* = 0.044) with the ΔOI_S18P_ (Appendix A). We further analyzed the data, classifying the patients into those who responded or not, according to the change seen in the Cardiac Index (CI) between baseline SR_BAS_ and 1hProne. There were 8 (42%) responders, who manifested an increase of 1.05 ± 0.6 L/min/m^2^ in the CI, and 12 (67%) non-responders, with a change of −0.8 ± 0.2 in the CI. Responders and non-responders presented similar responses in terms of the oxygenation indices or $\dot{Q}$s/$\dot{Q}$t following placement in prone position.

## 4. Discussion

In the present prospective study in ARDS patients ventilated with lung-protective ventilation, our main observations were: (a) the change in the Oxygenation Index between supine and 18 h prone (ΔOI_S18P_) was highly correlated to the respective change in the shunt fraction, to the change in the Oxygenation Index between supine and 1h prone and to the supine ARDS score; (b) the absolute value of the Oxygenation Index at 18 h prone was related to the concomitant shunt fraction and ARDS score and, to a lesser extent, to several other clinical variables at the same time point (PaCO_2_, respiratory system compliance and transpulmonary arterial pressure gradient among them); (c) patients with a positive increase in the Cardiac Index following prone positioning (responders) presented no significant change in any of the oxygenation indices used or in the shunt fraction compared to non-responders. Importantly, PVR and the RV function did not change, despite the significant improvement in the Oxygenation Index, the $\dot{Q}$s/$\dot{Q}$t and the other oxygenation parameters between baseline, 1h and 18h of pronation.

In our investigation, the improvement in the Oxygenation Index between supine and prone posture and the absolute value of the Oxygenation Index at 18 h prone were significantly associated with the simultaneous change and absolute value of the shunt fraction, respectively. Shunt physiology and a low ventilation/perfusion ratio ($\dot{V}$/$\dot{Q}$) are components of venous admixture and a major cause of hypoxemia in ARDS. It has been also demonstrated that ARDS is associated with the loss of surfactant function, while in supine posture, certain lung areas, mainly in the posterior diaphragmatic area [7], are compressed by the heart, creating low $\dot{V}$/$\dot{Q}$ conditions [5,7]. Clinical data suggest that, in ARDS lungs, those dorsal, collapsed, low-ventilated lung regions are more likely to remain atelectatic throughout the respiratory cycle [15]. More resent research, using electrical impedance tomography and dual energy CT scans, has clearly shown that prone position can reverse this pathophysiology because the low $\dot{V}$/$\dot{Q}$ lung regions present the greatest improvement with the rotation from the supine to the prone posture [16,17]. Albert et al. [6] demonstrated in an animal model that the shunt fraction decreased from a mean of 27.5% to 13.15% from the supine to the prone posture. In our clinical study, the $\dot{Q}$s/$\dot{Q}$t decreased from a mean of 30.1% to 23.0%, and this change correlated significantly with the respective changes in the PO_2_/FiO_2_ (r = 0.75, *p* = 0.0001) and in the OI (r = 0.62, *p* = 0.007). We underline that our results are derived from pulmonary artery catheterization (PAC), which may reflect global changes in V/Q mismatch. The Q_S_/Qt PAC values correlate significantly with regional V/Q changes obtained by electrical impedance tomography and dual energy CT [17]. In this respect, our findings, which reflect global V/Q improvement, are in line with the results provided by other methods assessing regional aeration and regional blood flow changes in the injured lung during pronation [16,17].

In the past decades clinical studies [10,18] have reported changes between supine and prone positions in patients who remained in the prone position for short time periods. The COVID-19 era has contributed to a change in the usual duration of pronation. To the best of our knowledge, our study is the first to report $\dot{Q}$s/$\dot{Q}$t and oxygenation changes after long-lasting pronation in non-COVID-19 ARDS patients. The association between the ΔOI_S18P_ and Δ$\dot{Q}$s/$\dot{Q}$t provides further clinical grounds to the current understanding that shunt reduction is a major mechanism for the improvement in oxygenation seen in the prone position.

Moreover, our study shows that this decrease in the $\dot{Q}$s/$\dot{Q}$t correlated significantly with the corresponding changes in respiratory system compliance (Figure 3) and the changes in driving pressure between postures. This suggests that prone position probably reduced shunt by increasing alveolar ventilation ($\dot{V}$_A_), most likely by ‘opening’ lung regions which were poorly ventilated (atelectatic) in the supine posture. This is in line with clinical data reporting less gravitationally induced pleural pressure gradient and, therefore, less inhomogeneity in the prone position compared to the supine [19,20], as well as with experimental [5] and clinical [16] data reporting that, while ventilation improves in the dorsal lung, blood flow remains greatest in this area and, hence, prone posture improves ventilation–perfusion matching by decreasing venous admixture [21].

One might argue that, since shunt physiology is a component of venous admixture, prone position could improve arterial oxygenation by an increase in the mixed venous O_2_ content. Taking into account that there was no significant difference in hemoglobin concentration levels and O_2_ consumption between postures in our patients, any change in mixed venous O_2_ content could be attributed to cardiac output changes. Indeed, we found a significant correlation between those two variables (data not shown). However, neither of the two presented any association with the oxygenation changes between supine and prone posture.

We acknowledge that our study was performed in a relatively small population, ARDS etiology was variable and the majority of patients had sepsis/septic shock. Moreover, procedures may be different from other settings (i.e., criteria for use of PAC, cardiac ultrasound). These points have to be taken into consideration when extrapolating our results to a general ICU population. On the basis that our population consisted of many septic patients, we cannot exclude that different stages of sepsis pathophysiology, concomitant vasodilation of the pulmonary vasculature and/or clinical therapies might have variable effects on venous oxygen saturation and cardiac output that might have obscured changes between postures.

One study demonstrated increased cardiac output in prone posture compared to supine but only in a proportion of patients [8]. Subjects who presented an increase in cardiac output did not present a concomitant increase in mixed venous O_2_. On this basis, considering that CO response to prone positioning may be variable, we sought to analyze our results by hypothesizing that some patients manifest an increase in cardiac output secondary to the position change from supine to prone (responders). We found that there were patients who presented a positive CO response early following prone positioning. We also found a correlation (r = 0.48 *p* = 0.044) between ΔCI_S1P_ and ΔOI_S18P_ but not between ΔCI_S1P_ and Δ$\dot{Q}$s/$\dot{Q}$t_S18P_ (Appendix A). This is in agreement with previous investigations which suggested that cardiac output decrease may be associated with improvement in $\dot{Q}$s/$\dot{Q}$t and, hence, in oxygenation [22]. A plausible explanation for such an association might be decreased perfusion of low-ventilated lung regions and, thus, decreased V/Q mismatch globally. We speculate that the absence of a significant association between ΔCI_S1P_ and Δ$\dot{Q}$s/$\dot{Q}$t_S18P_ in the present study could be the fact that the ΔQ_S_/Q_T_ (and ΔOI) may be affected by multiple factors and not only by the changes in the CI. Another explanation is that our population was too small to depict such associations. Hence, we assume that the oxygenation improvement in prone posture in ARDS patients with sepsis is less likely to be driven by a change in cardiac output, and, consequently, by the perfusion of the lungs, a hypothesis which is in line with previous observations [23,24,25].

In the present study, we sought to evaluate whether baseline hemodynamic or respiratory indices could predict changes in the OI following prone positioning. Clinical data on this are very limited. Pelosi et al. [10] reported that respiratory system changes (particularly Cst,w) may play a role in determining the oxygenation response in the prone position, whereas Blanch et al. [21] reported that PO_2_ and PCO_2_ may be valuable in identifying responders. However, patients were assessed for a relatively short period (60–120 min) in those studies. In our study, we assessed hemodynamic and respiratory variables after 18 h in the prone position. We found that ΔOI_S18P_ significantly correlated to several variables, but most importantly to the respective change in the shunt fraction, to the early change (1st hour) in the OI following prone positioning and to the severity of disease (ARDS score). Similarly, the absolute value of the Oxygenation Index following 18 h of pronation was related to the shunt fraction, the ARDS score at that time point and, to a lesser extent, to other variables (including PCO_2_ and respiratory system compliance) (Appendix A). These results underline the importance of the early PO_2_ response to the prone maneuver and of the reduction in the shunt fraction via probable alveolar recruitment induced by prone posture.

## 5. Conclusions

The present investigation provides physiologic and clinical data which suggest that oxygenation improvement following prone position in ARDS patients with sepsis is driven by the shunt fraction reduction and not by changes in hemodynamics. The relationship between changes in the OI in the first hour of pronation and changes in the OI after 18 h of pronation might be useful in the clinical practice in identifying patients who will present the best response to oxygenation and to prone position in the long term.

## Figures and Tables

**Figure 1 jcm-12-00760-f001:**
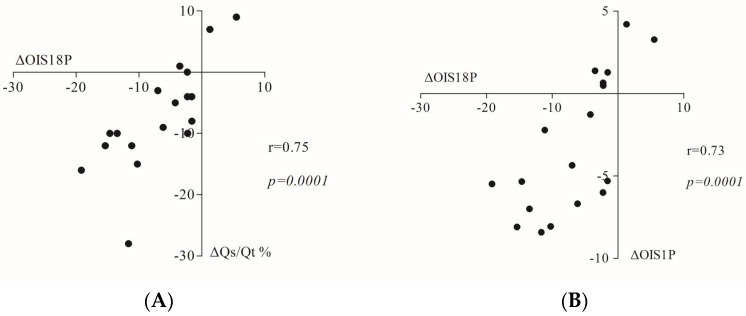
Pearson’s correlation coefficient r and *p*-value (*p*) between: (**A**) the change in the Oxygenation Index from baseline semi recumbent (SR_BAS_) to 18 h prone (ΔOI_S18hP_) and the respective change in shunt fraction (Δ$\dot{Q}$s/$\dot{Q}$t); (**B**) change in the Oxygenation Index from SR_BAS_ to 18 h prone (ΔOI_S18hP_) and the change in the Oxygenation Index from the SR_BAS_ to 1 h prone (ΔOI_S1hP_).

**Figure 2 jcm-12-00760-f002:**
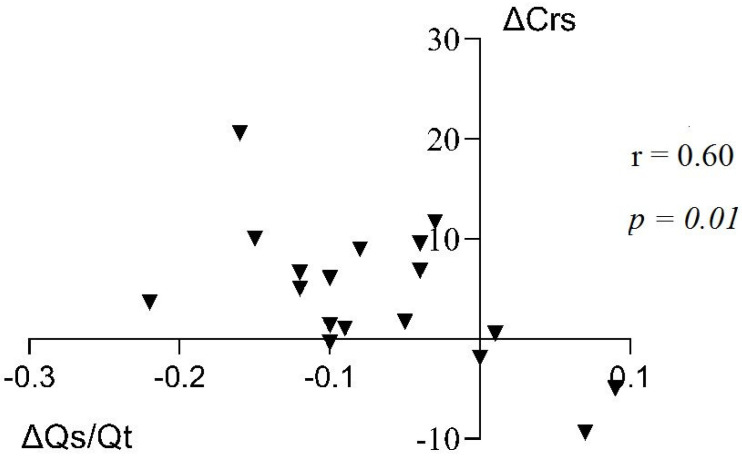
Pearson’s correlation coefficient (r) and *p*-value (*p*) between the change in shunt fraction (Δ$\dot{Q}$s/$\dot{Q}$t) and the synchronous change in the compliance of the respiratory system (ΔCrs, ml/mbar).

**Figure 3 jcm-12-00760-f003:**
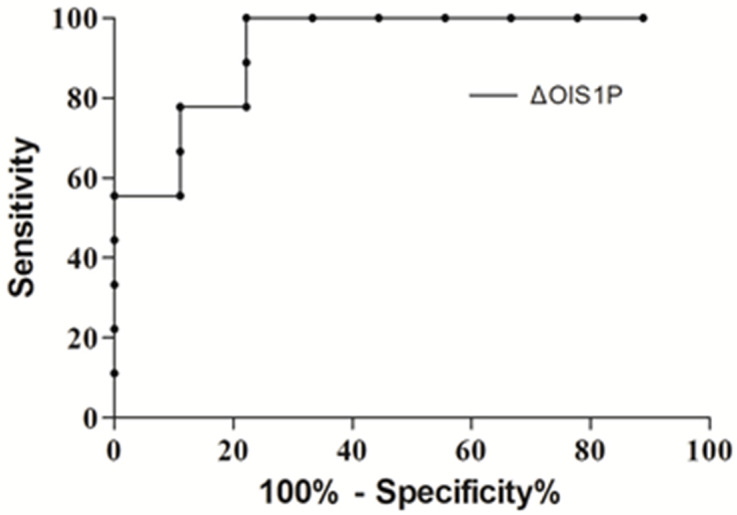
Receiver operator curve for the change in the Oxygenation Index from baseline semi recumbent (SR_BAS_) to 18hour prone (ΔOI_S18hP_) and the change in the Oxygenation Index from baseline semi recumbent (SR_BAS_) to 1 h prone (ΔOI_S1hP_).

**Table 1 jcm-12-00760-t001:** Baseline characteristics of participants.

Age, Years	62.8 ± 7.4
Male gender, n (%)	13 (72.2)
BSA, kg/m^2^	1.91 ± 0.26
APACHE II score at admission	17.3 ± 4.3
SOFA score at inclusion	9.34 ± 3.16
PaO_2_/FiO_2_ ratio at inclusion, mmHg	102.3 ± 29.4
Oxygenation Index at inclusion	20.6 ± 7.12
Lung Injury Score at inclusion	2.69 ± 0.54
PEEP at inclusion, mbar	9.8 ± 2.17
Sepsis, n (%)	16 (88.8%)
Septic shock, n (%)	5 (27.77%)
Pneumonia, n (%)	11 (61.1%)
Multiple Trauma n (%)	2 (11.11%)

Data are presented as mean ± SD, unless otherwise indicated. Pneumonia includes bacterial and viral pneumonia and pneumonia due to aspiration. Definition of abbreviations: BSA = Body surface area; APACHE = acute physiology and chronic health evaluation; PEEP = positive end-expiratory pressure, PaO_2_/FiO_2_ = the ratio of arterial oxygen partial pressure to the fraction of inspired oxygen SOFA = sequential organ failure assessment.

**Table 2 jcm-12-00760-t002:** Hemodynamic and pulmonary function data at baseline semi recumbent (SR_BAS_) posture, at 1 h prone posture (1hProne) and at 18 h prone posture (18hProne).

	SR_BAS_	1hProne	18hProne
NA drip, μg/kg/min	0.26 ± 0.28	0.25 ± 0.24	0.24 ± 0.34
Heart rate, bpm	97.59 ± 18.49	99.1 ± 18.80	100.39 ± 15.93
SAP, mmHg	128 ± 17.5	129 ± 15.5	128.1 ± 16.4
MAP, mmHg	79.23 ± 9.64	80.7 ± 12.08	79.66 ± 8.25
DAP, mmHg	54.94 ± 8.31	58.5 ± 9.81 *	57.88 ± 8.14
MPAP, mmHg	33.73 ± 7.49	34.58 ± 6.70	35.02 ± 1.63
CVP, mmHg	12.18 ± 4.57	12.76 ± 3.41	12.81 ± 4.66
PAOP, mmHg	14.18 ± 3.59	15.88 ± 3.42 *	15.39 ± 4.57
DPG, mmHg	7.41 ± 4.87	7.77 ± 3.90	9.72 ± 3.99
TPG, mmHg	19.58 ± 5.13	18.77 ± 5.29	19.63 ± 3.92
EDV, mL	233.76 ± 51.19	235.35 ± 53.63	221.59 ± 56.77
SV, mL/b	87.12 ± 18.02	91.47 ± 21.98	89.39 ± 27.36
EF %	38.56 ± 5.85	38.44 ± 6.13	39.61 ± 5.98
CO, L/min	8.69 ± 2.53	8.84 ± 2.06	8.79 ± 2.73
SVR, dyn × s/cm^5^	676.06 ± 243.39	668.19 ± 294.4	665.9 ± 298.71
PVR, dyn × s/cm^5^	189.93 ± 48.55	182.34 ± 46.5	194.48 ± 55.98
PEEP, mbar	10.18 ± 1.84	10.06 ± 1.78	9.77 ± 1.80
Pmean, mbar	19.41 ± 2.29	19.41 ± 2.06	19.55 ± 0.75
Tidal volume, mL	473.24 ± 79.15	492.9 ± 81.85	512.22 ± 84.96
C_rs_, mL/mbar	25.3 ± 7.0	28.9 ± 7.0	29.9 ± 7.50 *
PO_2_/FiO_2,_ mmHg	110.3 ± 45.69	141.0 ± 70.95 *	168.8 ± 63.98 *
OI	20.2 ± 7.56	16.59 ± 6.81	13.11 ± 5.57 *
$\dot{Q}$s/$\dot{Q}$t	0.30 ± 0.09	0.27 ± 0.08	0.23 ± 0.02 *
SaO_2,_%	94.18 ± 0.03	94.5 ± 0.03	96.17 ± 0.00 *
CaO_2_, mL/dL	14.56 ± 2.65	14.5 ± 3.01	14.61 ± 0.55
PaO_2_, mmHg	84.06 ± 16.79	102.94 ± 39.52 *	106.32 ± 6.62 *
pH arterial	7.31 ± 0.07	7.28 ± 0.1	7.33 ± 0.02
PCO_2_, mmHg	56.77 ± 11.78	59.65 ± 21 *	52.56 ± 3.23
DO_2_, mL/min	1281 ± 408.18	1273 ± 410.29	1327 ± 512.91
VO_2_, mL/min %	303 ± 115.75	299 ± 115.95	324 ± 143.84
P(v-a)CO_2_/C(a-v)O_2_	1.04 ± 0.91	1.5 ± 1.61	1.3 ± 0.78
SvO_2_,%	71.59 ± 0.05	73.1 ± 0.07	72.72 ± 0.05
CvO_2_, mL/dL	10.88 ± 2.21	11.01 ± 2.71	10.98 ± 1.96

Data are presented as mean ± SD, unless otherwise indicated. * *p* < 0.05 for the difference in respect to the SRBAS value. Definition of abbreviations: CaO_2_ = arterial oxygen content; CO = cardiac output; Crs = respiratory system compliance; CvO_2_ = mixed venous oxygen content; CVP = right atrial pressure; DAP = diastolic arterial pressure; SAP = systolic arterial pressure; MAP = mean arterial pressure; DPAP = diastolic pulmonary artery pressure; DPG = diastolic pulmonary vascular pressure gradient (DPAP-PAOP); EDV = end-diastolic volume of the right ventricle; EF = ejection fraction (of right ventricle); FiO_2_ = fraction of inspired Oxygen; HR = heart rate; MAP = mean arterial pressure; MPAP = mean pulmonary artery pressure; NA = Noradrenaline OI = Oxygenation Index = FiO_2_ × Mean Airway Pressure/PO_2_; PAOP = pulmonary artery occlusion pressure; PaO_2_ = Pressure of arterial Oxygen; PCO_2_ = arterial carbon dioxide partial pressure; PEEP = positive end-expiratory pressure; Pmean = mean airway pressure; PVR = pulmonary vascular resistance. $\dot{Q}$s/$\dot{Q}$t = fraction of pulmonary shunt to cardiac output; SaO_2_ = arterial hemoglobin saturation; SPAP = systolic pulmonary artery pressure; SV = stroke volume (right ventricle); SvO_2_ = mixed venous hemoglobin saturation; SVR = systemic vascular resistance; TPG = transpulmonary gradient = mean pulmonary artery pressure - pulmonary artery occlusion pressure (MPAP-PAOP).

**Table 3 jcm-12-00760-t003:** Significant relationships between the change in the Oxygenation Index from baseline semi recumbent posture to 18 h prone posture (ΔOI_S18hP_) and clinical, hemodynamic and respiratory indices.

	Correlation Coefficient *	*p* Value
ARDS score at baseline	−0.73	0.009
Δ$\dot{Q}$s/$\dot{Q}$t from baseline to 1hProne	0.55	0.018
ΔPaCO_2_ from baseline to 1hProne	0.47	0.045
Δ$\dot{Q}$s/$\dot{Q}$t from baseline to 18hProne	0.75	0.000
ΔCrs from baseline to 18hProne	−0.60	0.01

* Pearson’s r, unless otherwise indicated. Definition of abbreviations: ARDS = acute respiratory distress syndrome; Δ = Changes; OI = Oxygenation Index; PCaO_2_ = carbon dioxide arterial partial pressure; $\dot{Q}$s/$\dot{Q}$t = fraction of pulmonary shunt to cardiac output; Crs = compliance respiratory system.

## Data Availability

The datasets used and analyzed during the current study are available from the corresponding author on reasonable request.

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
