# Peer review of "Hemodynamic and Respiratory Changes following Prone Position in Acute Respiratory Distress Syndrome Patients: A Clinical Study"

_jcm, 2023, doi:10.3390/jcm12030760_

Round 1
Reviewer 1 Report
I read with interest the manuscript entitled “Hemodynamic and respiratory changes following prone position in Acute Respiratory Distress Syndrome patients: a clinical study”. The authors evaluated parameters associated to oxygenation improvement in patients undergoing prone positioning and found that oxygenation improvement is associated to a reduction of shunt. This is not new data, since several study based on CT scan and Electrical impedance tomography (which are not discussed by the authors) showed already this result. Nevertheless, the use of invasive hemodynamic monitoring is interesting, the data are well presented and the paper is well written and provides interesting results. I still have few comments:
GENERAL COMMENTS
- The result found by the authors have been previously evaluated using other techniques, like Electrical impedance tomography V/Q mismatch calculation and dual energy CT scan. Please discuss your results taking into account this strong background.
- what the authors evaluated is a global change in V/Q mismatch. Other technologies, like CT scan and EIT provided regional information about changes in V/Q mismatch in patients undergoing Prone Position. Please discuss this point.
- I am not totally sure about the calculations of correlation coefficients since many couples variables seems to be not independent (derived from the same variables). Please check the correlation table (DOI and P/F ratio)
- You did not evaluate a general ARDS population but a population of septic patients with ARDS, hemodynamic instability, high doses of vasopressors and severely depressed SVR. You should interpretate your data considering this and you cannot generally talk about ARDS patients.
- I find interesting that only patients that did not increase CI after prone positioning improved shunt and OI. Please can the author add some more comments about this data in the discussion? How do they interpretate this data?
SPECIFIC COMMENTS
- line 82: explain in details what conventional mechanical ventilation means and how was set mechanical ventilation both in supine and in prone position. Specifically, explain how PEEP setting was conducted since there is still poor consent on it. Specifically, there is to date no evidence about PEEP titration on the basis of preload assessment (add reference).
- was supination after PP based on any threshold? E.g. oxygenation threshold?
- did you calculate a sample size for your study? If not, please explain why, since there is literature about oxygenation index in ARDS. This may be a limitation of the study and must be reported.
- your patients are almost all septic under high doses of vasopressors (mean 0.26 but way higher, considering the standard deviation). Do you think that the hemodynamic assessment in this population may reflect a general ARDS population? I have some doubts considering the possible effects . We know hemodynamic (and probably V/Q mismatch) is different in septic patients.
- SAP is missing in the table 2. Please consider adding this information.
- table 3: I don’t understand the correlation between DOIs18hP , OI and DOI. First, you should declare at which time OI and DOI refers to (not reported). Second, probably the correlation is wrong since the assumption of a correlation calculation is the independence of the variables. The same applies for PaO2/FiO2 ratios reported in the table, since the calculation of both OI and PaO2/FiO2 ratio contains variables derived from the same data (FiO2 and PaO2).
- line 142: please declare when Oxygenation Index was measured.
Author Response
We were pleased to hear that the JCM is interested in a revised version of our manuscript entitled “Hemodynamic and respiratory changes following prone position in Acute Respiratory Distress Syndrome patients: a clinical study.” by M Baka et al.
We are grateful to the reviewers for their helpful suggestions and comments. All comments have been addressed in the revised version. Having made this revision, we hope that the manuscript would be acceptable for publication at the JCM in its current version
REVIEWER 1.
I READ WITH INTEREST THE MANUSCRIPT ENTITLED “HEMODYNAMIC AND RESPIRATORY CHANGES FOLLOWING PRONE POSITION IN ACUTE RESPIRATORY DISTRESS SYNDROME PATIENTS: A CLINICAL STUDY”. THE AUTHORS EVALUATED PARAMETERS ASSOCIATED TO OXYGENATION IMPROVEMENT IN PATIENTS UNDERGOING PRONE POSITIONING AND FOUND THAT OXYGENATION IMPROVEMENT IS ASSOCIATED TO A REDUCTION OF SHUNT. THIS IS NOT NEW DATA, SINCE SEVERAL STUDY BASED ON CT SCAN AND ELECTRICAL IMPEDANCE TOMOGRAPHY (WHICH ARE NOT DISCUSSED BY THE AUTHORS) SHOWED ALREADY THIS RESULT. NEVERTHELESS, THE USE OF INVASIVE HEMODYNAMIC MONITORING IS INTERESTING, THE DATA ARE WELL PRESENTED AND THE PAPER IS WELL WRITTEN AND PROVIDES INTERESTING RESULTS. I STILL HAVE FEW COMMENTS:
We thank you for your thorough review that helped us improve our manuscript
GENERAL COMMENTS
- 1. THE RESULT FOUND BY THE AUTHORS HAVE BEEN PREVIOUSLY EVALUATED USING OTHER TECHNIQUES, LIKE ELECTRICAL IMPEDANCE TOMOGRAPHY V/Q MISMATCH CALCULATION AND DUAL ENERGY CT SCAN. PLEASE DISCUSS YOUR RESULTS TAKING INTO ACCOUNT THIS STRONG BACKGROUND.
Thank you for your comment. Indeed, modern technology has brought new insight to an old issue. We regret not having discussed our findings in relation to this strong background. In light of your comment, we have added relevant comments in the discussion section (page 8, lines 199-202 and 207-212) and references (15,16,17).
- 2. WHAT THE AUTHORS EVALUATED IS A GLOBAL CHANGE IN V/Q MISMATCH. OTHER TECHNOLOGIES, LIKE CT SCAN AND EIT PROVIDED REGIONAL INFORMATION ABOUT CHANGES IN V/Q MISMATCH IN PATIENTS UNDERGOING PRONE POSITION. PLEASE DISCUSS THIS POINT.
Thank you for your comment. We agree and we have added a relevant comment in discussion acknowledging your remark discussion (page 8, lines 199-202 and 207-212) and references.
-3. I AM NOT TOTALLY SURE ABOUT THE CALCULATIONS OF CORRELATION COEFFICIENTS SINCE MANY COUPLES VARIABLES SEEMS TO BE NOT INDEPENDENT (DERIVED FROM THE SAME VARIABLES). PLEASE CHECK THE CORRELATION TABLE (DOI AND P/F RATIO)
We thank the reviewer for this remark. In light of your comment to avoid adding confusion we have now omitted OI, PO2/FiO2 from table3.
- 4. YOU DID NOT EVALUATE A GENERAL ARDS POPULATION BUT A POPULATION OF SEPTIC PATIENTS WITH ARDS, HEMODYNAMIC INSTABILITY, HIGH DOSES OF VASOPRESSORS AND SEVERELY DEPRESSED SVR. YOU SHOULD INTERPRETATE YOUR DATA CONSIDERING THIS AND YOU CANNOT GENERALLY TALK ABOUT ARDS PATIENTS.
You are right. We recognize that septic patients in our study do not represent the general ARDS population. A paragraph in the discussion section (page:8-9, line238-246) underlines the limitations in the interpretation of our results.
-5. I FIND INTERESTING THAT ONLY PATIENTS THAT DID NOT INCREASE CI AFTER τPRONE POSITIONING IMPROVED SHUNT AND OI. PLEASE CAN THE AUTHOR ADD SOME MORE COMMENTS ABOUT THIS DATA IN THE DISCUSSION? HOW DO THEY INTERPRETATE THIS DATA?
Thank you for your comment. In our study we found a correlation (r=0.48 p=0.044) between ΔΟΙ [from SRBAS to 18h prone] and ΔCI [from SRBAS to 1st h prone] (Supplement Table). However, the ΔQ̇s/Q̇t [from baseline to 18hProne] between CI responders and non-responders were –0.048 vs –0.086 (p>0.05); similar results were found for the ΔOI [from baseline to 18hProne] (-6,3 vs -6.8). We speculate that the absence of significant difference in the absοlute values of ΔΟΙ and ΔQ̇s/Q̇t between responders and non-responders could be due to the fact that ΔQ̇s/Q̇t and ΔΟΙ are affected by multiple factors and not only by the changes in CI or due to the fact that our population was small to depict such differences. In light of your comment, we clarify this points in results and discussion (page7 line170-171. And page 9 lines256-268)
SPECIFIC COMMENTS
- 6. LINE 82: EXPLAIN IN DETAILS WHAT CONVENTIONAL MECHANICAL VENTILATION MEANS AND HOW WAS SET MECHANICAL VENTILATION BOTH IN SUPINE AND IN PRONE POSITION. SPECIFICALLY, EXPLAIN HOW PEEP SETTING WAS CONDUCTED SINCE THERE IS STILL POOR CONSENT ON IT. SPECIFICALLY, THERE IS TO DATE NO EVIDENCE ABOUT PEEP TITRATION ON THE BASIS OF PRELOAD ASSESSMENT (ADD REFERENCE).
You are right that this point needs clarification. In light of your remark, further information is now provided on the settings session in the text (page:2 line 85-94). Additionally, we clarify that the purpose of cardiac ultrasound examination was to obtain information during PEEP optimization (based on open lung strategies) to avoid acute cor pulmonale [ref Baron]. Our physicians were advised to use ultrasound data; however there was no specific PEEP maneuvers based on certain Ultrasound derived targets. We regret the confusion created on an effort to present settings in a shortly manner. In light of your comment, we added a clarification in Methods (page lines)
-7.WAS SUPINATION AFTER PP BASED ON ANY THRESHOLD? E.G. OXYGENATION THRESHOLD?
Thank you for your remark. Supination after PP was performed in a timely manner, i.e., depending on the time of staff shift change and provided the patient was prone for at least 16hours. Please find relevant information on the settings section ( p:2-3, line 95-98)
. -8.DID YOU CALCULATE A SAMPLE SIZE FOR YOUR STUDY? IF NOT, PLEASE EXPLAIN WHY, SINCE THERE IS LITERATURE ABOUT OXYGENATION INDEX IN ARDS. THIS MAY BE A LIMITATION OF THE STUDY AND MUST BE REPORTED.
Thank you for your comment. We have not performed a power sample size calculation in advance in this observational study. Based on available literature (Leonard Go CCM 2017) OI in the large ARDS trials presents overall a change of 1.6 units (from 12.8 to 14.4) at the first 7 days and approximately of 0.5 (SD=11) in the first 24 hours. On this basis the change in OI in our study of 7.1 (from 20.2 to 13.1) that was observed in our population (n=18) gives a power of 0.8, alpha 0.05. In light of your remark, we have added a relevant comment in discussion underlining the small size of our population (page:8-9 lines: 238-246).
-9. YOUR PATIENTS ARE ALMOST ALL SEPTIC UNDER HIGH DOSES OF VASOPRESSORS (MEAN 0.26 BUT WAY HIGHER, CONSIDERING THE STANDARD DEVIATION). DO YOU THINK THAT THE HEMODYNAMIC ASSESSMENT IN THIS POPULATION MAY REFLECT A GENERAL ARDS POPULATION? I HAVE SOME DOUBTS CONSIDERING THE POSSIBLE EFFECTS. WE KNOW HEMODYNAMIC (AND PROBABLY V/Q MISMATCH) IS DIFFERENT IN SEPTIC PATIENTS.
Thank you for your comment. We agree that the interpretation of our results should take into account the point you have underlined. This point is now acknowledged as a limitation of our study. (Discussion section page:8-9 lines: 238-246).
- 10. SAP IS MISSING IN THE TABLE 2. PLEASE CONSIDER ADDING THIS INFORMATION.
Thank you for your comment. Done as you recommended. (Table 2, page 4).
- 11. TABLE 3: I DON’T UNDERSTAND THE CORRELATION BETWEEN DOIS18HP , OI AND DOI. FIRST, YOU SHOULD DECLARE AT WHICH TIME OI AND DOI REFERS TO (NOT REPORTED). SECOND, PROBABLY THE CORRELATION IS WRONG SINCE THE ASSUMPTION OF A CORRELATION CALCULATION IS THE INDEPENDENCE OF THE VARIABLES. THE SAME APPLIES FOR PAO2/FIO2 RATIOS REPORTED IN THE TABLE, SINCE THE CALCULATION OF BOTH OI AND PAO2/FIO2 RATIO CONTAINS VARIABLES DERIVED FROM THE SAME DATA (FIO2 AND PAO2).
We thank the reviewer for this remark. We agree that the presentation of indices derived from the same variable (i.e. PO2) may lead in autocorrelation of variables and can be possibly misleading. In light of your previous comment, we have omitted those values from table 3 and we provide definition for each variable.
- 12. LINE 142: PLEASE DECLARE WHEN OXYGENATION INDEX WAS MEASURED.
You are right, we added this information (at SRbas: semi recumbent base line) as asked.

Reviewer 2 Report
I read with great interest the manuscript entitled “Hemodynamic and respiratory changes following prone position in Acute Respiratory Distress Syndrome patients: a clinical study” by Baka et al. The authors aimed to assess oxygenation and hemodynamic changes following prone ventilation in patients with ARDS and identify parameters associated with improved oxygenation. Pulmonary artery catheter parameters were recorded for 18 study patients with ARDS undergoing protective ventilation. Recording was performed in supine position, and at 1h and 18h post initiation of prone ventilation. Authors concluded that oxygenation improvement with prone ventilation was correlated with shunt fraction reduction and not by changes in hemodynamics. Please find my comments below.
- The criteria used to drive the decision to prone does not appear to be standardized – was this entirely determined by the treatment team? This has potential for inducing bias. Standardization of the criteria used to determine which patients were subjected to prone ventilation would improve strength (i.e. PaO2/FiO2 ratio <150 on >60% FiO2/PEEP5 similar to PROSEVA).
- Similar to above- what criteria were utilized to determine which patients underwent pulmonary artery catheter placement? If determined entirely by the treatment team, this creates a potential for bias also.
- Authors indicated that PEEP titration was performed with the aid of echocardiographic parameters. Was this standardized or also left to the discretion of the treatment team? The lack of standard PEEP titration also has potential for impacting the findings.
- Etiology of ARDS was variable. Patients with septic shock were likely on vasoactive agents which may have impacted the hemodynamic findings.
- Study population/sample size is small – only 18 patients, which has the potential for increasing bias and may prevent the findings from being extrapolated
Though the above issues in the investigation clearly exist, I do find the manuscript contributes to the literature by providing physiologic and clinical data that indicates that the oxygenation improvement from prone ventilation in ARDS is likely driven by shunt fraction reduction.
Author Response
We were pleased to hear that the JCM is interested in a revised version of our manuscript entitled “Hemodynamic and respiratory changes following prone position in Acute Respiratory Distress Syndrome patients: a clinical study.” by M Baka et al.
We are grateful to the reviewers for their helpful suggestions and comments. All comments have been addressed in the revised version. Having made this revision, we hope that the manuscript would be acceptable for publication at the JCM in its current version
REVIEWER 2
I READ WITH GREAT INTEREST THE MANUSCRIPT ENTITLED “HEMODYNAMIC AND RESPIRATORY CHANGES FOLLOWING PRONE POSITION IN ACUTE RESPIRATORY DISTRESS SYNDROME PATIENTS: A CLINICAL STUDY” BY BAKA ET AL. THE AUTHORS AIMED TO ASSESS OXYGENATION AND HEMODYNAMIC CHANGES FOLLOWING PRONE VENTILATION IN PATIENTS WITH ARDS AND IDENTIFY PARAMETERS ASSOCIATED WITH IMPROVED OXYGENATION. PULMONARY ARTERY CATHETER PARAMETERS WERE RECORDED FOR 18 STUDY PATIENTS WITH ARDS UNDERGOING PROTECTIVE VENTILATION. RECORDING WAS PERFORMED IN SUPINE POSITION, AND AT 1H AND 18H POST INITIATION OF PRONE VENTILATION. AUTHORS CONCLUDED THAT OXYGENATION IMPROVEMENT WITH PRONE VENTILATION WAS CORRELATED WITH SHUNT FRACTION REDUCTION AND NOT BY CHANGES IN HEMODYNAMICS. PLEASE FIND MY COMMENTS BELOW.
We thank you for your thorough review that helped us improve our manuscript
- 1.THE CRITERIA USED TO DRIVE THE DECISION TO PRONE DOES NOT APPEAR TO BE STANDARDIZED – WAS THIS ENTIRELY DETERMINED BY THE TREATMENT TEAM? THIS HAS POTENTIAL FOR INDUCING BIAS. STANDARDIZATION OF THE CRITERIA USED TO DETERMINE WHICH PATIENTS WERE SUBJECTED TO PRONE VENTILATION WOULD IMPROVE STRENGTH (I.E. PAO2/FIO2 RATIO <150 ON >60% FIO2/PEEP5 SIMILAR TO PROSEVA).
Thank you for your comment. Criteria for placement in prone position were similar to the PROSEVA trial and are now described in Methods. (page:2 line:67)
- SIMILAR TO ABOVE- WHAT CRITERIA WERE UTILIZED TO DETERMINE WHICH PATIENTS UNDERWENT PULMONARY ARTERY CATHETER PLACEMENT? IF DETERMINED ENTIRELY BY THE TREATMENT TEAM, THIS CREATES A POTENTIAL FOR BIAS ALSO.
Thank you for your comment. Decisions regarding the placement of PAC were taken by the treating physicians whereas all cases were discussed in a daily multidisciplinary meeting. We acknowledge that the absence of strict standardized criteria for PAC may be a limitation for interpolating our findings to other settings. However, we aimed to study ARDS patients under conditions that can be met in the everyday day clinical practice and decisions were based as explained before, on treating physicians and daily multidisciplinary meeting. We acknowledge this point as a potential limitation in the discussion section. (Page: 8-9, lines:238-246)
- AUTHORS INDICATED THAT PEEP TITRATION WAS PERFORMED WITH THE AID OF ECHOCARDIOGRAPHIC PARAMETERS. WAS THIS STANDARDIZED OR ALSO LEFT TO THE DISCRETION OF THE TREATMENT TEAM? THE LACK OF STANDARD PEEP TITRATION ALSO HAS POTENTIAL FOR IMPACTING THE FINDINGS.
You are right that this point needs clarification. We clarify that the purpose of cardiac ultrasound examination was to obtain information during PEEP optimization (based on open lung strategy) to avoid acute cor pulmonale [ref 9]. Our physicians were advised to use ultrasound data; however, there was no specific PEEP maneuvers based on certain Ultrasound derived targets. We regret the confusion created on an effort to present settings in a shortly manner. In light of your comment, we added a clarification in settings section (page:2 lines: 85-94)
- ETIOLOGY OF ARDS WAS VARIABLE. PATIENTS WITH SEPTIC SHOCK WERE LIKELY ON VASOACTIVE AGENTS WHICH MAY HAVE IMPACTED THE HEMODYNAMIC FINDINGS.
Thank you for your comment. We agree and we have underlined that our population consisted of many septic patients, we cannot exclude that different stages of sepsis pathophysiology, concomitant vasodilation of the pulmonary vasculature and/or clinical therapies might have variable effects on venous oxygen saturation and cardiac output that might have obscured changes between postures (page:8-9 lines:238-246).
- STUDY POPULATION/SAMPLE SIZE IS SMALL – ONLY 18 PATIENTS, WHICH HAS THE POTENTIAL FOR INCREASING BIAS AND MAY PREVENT THE FINDINGS FROM BEING EXTRAPOLATED.
You are certainly right that we have studies a relatively small population in this observational study. In light of your remark, we have added a relevant comment in discussion underling the small size of our population (page:8 line:238-246).
THOUGH THE ABOVE ISSUES IN THE INVESTIGATION CLEARLY EXIST, I DO FIND THE MANUSCRIPT CONTRIBUTES TO THE LITERATURE BY PROVIDING PHYSIOLOGIC AND CLINICAL DATA THAT INDICATES THAT THE OXYGENATION IMPROVEMENT FROM PRONE VENTILATION IN ARDS IS LIKELY DRIVEN BY SHUNT FRACTION REDUCTION.

Round 2
Reviewer 1 Report
I congratulate the authors for improving their manuscript and would like to thank for their precise answers. I think that the manuscript improved after the revision and I don't have further comments.
Reviewer 2 Report
My initial comments, questions and suggestions have been appropriately addressed by the authors.